# Immobile and mobile excitations of three-spin interactions on the diamond chain

**Maximilian Bayer**[⋆]**, Maximilian Vieweg**[°]**, and Kai Phillip Schmidt**[†]

1 Friedrich-Alexander-Universität Erlangen-Nürnberg (FAU), Department of Physics,
Staudtstraße 7, 91058 Erlangen, Germany

⋆ maximilian.bayer@fau.de , ∘max.vieweg@fau.de , † kai.phillip.schmidt@fau.de

## Abstract

**We present a solvable one-dimensional spin-1/2 model on the diamond chain featuring three-spin interactions, which displays both, mobile excitations driving a second-order phase transition between an ordered and a $\mathbb{Z}_2$-symmetry broken phase, as well as non-trivial fully immobile excitations. The model is motivated by the physics of fracton excitations, which only possess mobility in a reduced dimension compared to the full model. We provide an exact mapping of this model to an arbitrary number of independent transverse-field Ising chain segments with open boundary conditions. The number and lengths of these segments correspond directly to the number of immobile excitations and their respective distances from one another. Furthermore, we demonstrate that multiple immobile excitations exhibit Casimir-like forces between them, resulting in a non-trivial spectrum.**

## 1 Introduction

A usual assumption in the field of quantum many-body physics, when dealing with low-energy quasi-particle excitations is that they inherit their degrees of freedom from the lattice they are defined on. A class of excitations which in recent years gained a lot of interest are fractons and sub-dimensional particles, which defy the above notion. Such excitations are only mobile within a subsystem manifold that has a dimension smaller than that of the lattice. The notion fracton was coined in the context of exactly solvable topologically ordered quantum spin systems [1,2], some of which are candidates for stable quantum memory [3]. Associated fracton phases are typically classified into two categories: Type-I fracton phases, such as in the X-Cube model [2] and type-II fracton phases, such as in Haah's Code [3]. Both types of fracton phases are defined on three-dimensional lattices, are translation-invariant, possess an energy gap, exhibit subextensive ground-state degeneracy, display long-range entanglement in the ground state, and feature immobile topological point-like excitations known as fractons, which give these systems their name. The characteristic immobility of the fractons is robust against any local disturbances. For type-I fracton phases, fractons can still form composite excitations that can move in one (lineons), two (planons), or three dimensions. In contrast, such composite mobile excitations are not possible in type-II fracton phases. Since their initial introduction, the study of their dynamical properties has become a compelling field of research in its own right. Systems with fractons can, for example, exhibit intriguing hydrodynamic universality classes [4] and may violate the eigenstate thermalization hypothesis [5,6]. A comprehensive review of fractons can be found in Refs. [7–9]. The concept of fractons can be transferred to non-topological two-dimensional systems. Often, the non-trivial kinetic properties of fracton excitations arrive from the presence of subsystem symmetries [10–15]. Interestingly, recent

findings have established connections between the topologically ordered toric code and such fracton excitations [16–18].

In this work we investigate an integrable one-dimensional spin-1/2 model of three-spin interactions on the diamond chain. Besides regular mobile excitations, which can move along the one-dimensional extent of the system, this model features fully immobile excitations localized on a given site. Although these types of immobile excitations do not exhibit the typical characteristics of fracton excitations, such as gaining an additional degree of mobility when present in multiples, these excitations remain immobile even when an arbitrary number of them are excited because they are protected by exact quantum numbers. They are therefore an exciting zero-dimensional counterpart of fracton excitations in higher dimensions with respect to their reduced dimensionality.

A more accurate interpretation of the model and its immobile excitations actually arises from viewing it as a one-dimensional cut-out of a higher-dimensional model with fracton excitations. Indeed, one can imagine a higher-dimensional model with at least one fracton excitation, whose mobility is restricted (alone or in pairs) to a one-dimensional subsystem due to subsystem symmetries. If we know where to cut out a slice of this model that is orthogonal to the subsystem in which the fracton excitation was allowed to move, we are left with a system in which these excitations, which previously had a single degree of mobility, now lack any direction in which to move and consequently become immobile. Similarly, the one-dimensional subsystem symmetry will transition to a zero-dimensional local symmetry that precisely conserves the immobile excitations. Such a cut-out procedure is of course only possible if the interactions are sufficiently local and fit on the lower-dimensional cutout system.

Interestingly, an extensive number of local symmetries allows us to provide an exact mapping of our model to an arbitrary number of independent transverse-field Ising chain segments with open boundary conditions. The number and lengths of these segments correspond directly to the number of immobile excitations and their respective distances from one another. While fully localized, multiple immobile excitations exhibit Casimir-like forces between them, resulting in a non-trivial spectrum.

The article is organized as follows. In Sec. 2 we introduce the model including its symmetries, the immobile excitations, and the exact duality mapping in terms of pseudo-spin transverse-field Ising chains. The analytic solution of the low-energy properties is given in Sec. 3. In the subsequent Sec. 4 we discuss the quantum phase diagram of the model and we conclude in Sec. 5.

## 2   Model

In this section we define the model studied in this work, discuss its symmetries and the resulting mappings to the one-dimensional transverse field Ising model in its different symmetry sectors.

We consider a quasi-one-dimensional diamond chain with linear length $L$ which is built by alternating between single sites and dimers of spin-1/2 particles as illustrated in Fig. 1. The Hamiltonian is defined by three-spin interactions between the two spins of a dimer and a neighbouring single site. We further include a transverse field in $z-$direction acting on every spin. In appendix B we discuss the results obtain by introducing magnetic fields with differing strengths on the single and dimer sites.

The three-site unit cell of the diamond chain is given by the two spins $A$ and $B$ of a dimer and the neighbouring spin $C$. The number of unit cells in a finite version of this model is thus equal to the number of dimer sites, which we will call $N_{\mathrm{d}}$ and the linear length $L$ is given by $2N_{\mathrm{d}}$. Looking at a finite periodic chain with $N_{\mathrm{d}}$ dimers, the full Hamiltonian is given by

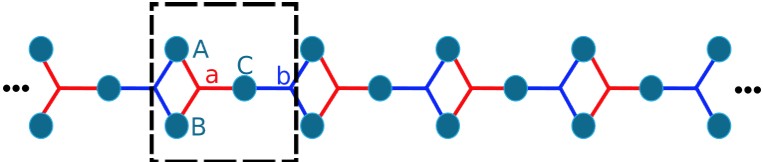

Figure 1: Diamond lattice with alternating single and dimer sites (blue dots). The three-spin interactions given by products of $\sigma^x$ operators are indicated by the red and blue three-particle bonds labelled $a, b$ respectively. The unit cell composed by sites $A$, $B$, and $C$ is illustrated as the dashed box.

$$H = J \sum_{i=1}^{N_d} \left( \sigma_{i,A}^x \sigma_{i,B}^x \sigma_{i,C}^x + \sigma_{i,C}^x \sigma_{i+1,A}^x \sigma_{i+1,B}^x \right) + h \sum_{i=1}^{N_d} \sum_{f \in \{A,B,C\}} \sigma_{i,f}^z \,, \tag{1}$$

with a three-spin coupling strength $J$ and a magnetic field strength $h$. Sums are taken over unit cells with periodic boundary conditions. The signs of $J$ and $h$ are taken to be positive. These signs are irrelevant for our purposes as flipping all spins in z(x)-direction using the operator $U_z = \prod_{i=1}^{N_d} \sigma_{i,A}^z \sigma_{i,B}^z \sigma_{i,C}^z$ ($U_x = \prod_{i=1}^{N_d} \sigma_{i,A}^x \sigma_{i,B}^x \sigma_{i,C}^x$) maps $J \to -J$ ($h \to -h$) and keeps the spectrum of the Hamiltonian $H$ unchanged

$$H(J,h)\,|\Psi\rangle = E\,|\Psi\rangle \iff H(-J,h)\,(U_z\,|\Psi\rangle) = E\,(U_z\,|\Psi\rangle) \tag{2}$$

and similar for $U_x$.

## 2.1 Symmetries

In order to discuss in Subsec. 2.4 the mapping to the transfers field Ising model and our models phases in the limiting cases we summaries its symmetries. This model has a significant number of subsystem symmetries - namely one for each dimer $i$. As the full system is one-dimensional, these local subsystems are zero-dimensional. These symmetries are given by

$$S_i = \sigma_{i,A}^z \sigma_{i,B}^z \tag{3}$$

with eigenvalue $s_i \in \{\pm 1\}$. In addition to these local symmetries there is a global symmetry similar to a total spin-flip symmetry. This symmetry flips one spin of every dimer and every spin between dimers

$$S_{\text{flip}} = \prod_{i=1}^{N_d} \sigma_{i,A}^z \sigma_{i,C}^z \tag{4}$$

with eigenvalue $s_{\text{flip}} \in \{\pm 1\}$. All other symmetry involving spin-flips can be written as a product of these two types of symmetries.

## 2.2 Limiting cases

The Hamiltonian (1) in the thermodynamic limit displays an ordered phase in the limit $h \gg J$ and a $\mathbb{Z}_2$ symmetry-broken phase in the limit $J \gg h$. The $\mathbb{Z}_2$ symmetry is given by $S_{\text{flip}}$. For $h = 0$, the degeneracy of the ground state is $2^{N_d+1}$ corresponding to the $N_d$ local symmetries $S_i$ and the global symmetry $S_{\text{flip}}$. In the range $h/J \ll 1$ second-order perturbation theory leads to an energy splitting between the two ground states

$$|\Psi_\pm\rangle = \left( \mathbb{1} \pm S_{\text{flip}} \right) \prod_{i=1}^{N_d} \left( \mathbb{1} + S_i \right) |\rightarrow\rangle \,, \tag{5}$$

where $|\rightarrow\rangle$ denotes the product state with all spins pointing in positive $x$-direction and all remaining previous ground states, leading to the $\mathbb{Z}^2$ symmetric phase (see Appendix C for the calculation).

The quantum phase transition is driven by the mobile modes of the model, which in the limit $h \gg J$ are adiabatically connected to a single spin-flip on a single site $C$ in a unit cell. Note that the quantum phase transition has to be driven by mobile excitations as the immobile excitations are fully local and can thus not lead to symmetry breaking as of Elitzur's theorem [19].

## 2.3   Immobile excitations

Taking a look at the low-energy excitations for perturbatively small $J$ we notice that a spin-flip between two dimers can travel freely to any other single site on the diamond chain just by applying the three-spin interactions accordingly, while a single spin-flip on a dimer cannot move to another dimer by any number of finitely many applications of the three-spin interactions and is thus immobile in any finite order in perturbative theory.

In this work we will calculate the energy of this immobile excitation analytically in the parameter range, where it is adiabatically connected to a single spin-flip on a dimer. To do so we have to identify the appropriate symmetry sectors. A single immobile excitation lives in the symmetry sector where all except one of the local subsystem symmetries $S_i$ have the eigenvalue $s_i = +1$, while the remaining one has the eigenvalue $-1$. The global symmetry has an eigenvalue of $s_{\text{flip}} = +1$ as does the ground-state sector of the $h \gg J$ limit. The ground-state energy in this sector will be equal to the ground-state energy in the sector without the immobile excitation plus its excitation energy. The symmetry sector without immobile mode is given by setting the eigenvalues of all symmetries to $+1$. We will therefore calculate the ground-state energies of the model in both of these sectors for finite $N_{\text{d}}$, calculate their difference and look at the limit $N_{\text{d}} \rightarrow \infty$ in order to calculate its energy in the thermodynamic limit. In order to perform these calculations we introduce duality mappings depending on the chosen symmetry sector.

## 2.4   Bijective mappings to transverse-field Ising models

We introduce a duality mapping from the sites of our model to the three spin bonds depicted in Fig. 1 by mapping the three-spin interactions to pseudo-spins $1/2$

$$
\begin{aligned}
\sigma_{i,A}^x \sigma_{i,B}^x \sigma_{i,C}^x &\rightarrow \tau_{i,a}^z \\
\sigma_{i,C}^x \sigma_{i+1,A}^x \sigma_{i+1,B}^x &\rightarrow \tau_{i,b}^z \, .
\end{aligned}
\tag{6}
$$

The index $i$, labels the unit cell, the second label $a/b$ distinguishes between a right/ left facing bond, depicted red/ blue in figure 1. Given a periodic chain with $N_{\text{d}}$ dimers its Hilbert space is $2^{3N_{\text{d}}}$-dimensional, while the Hilbert space of the pseudo-spins is only $2^{2N_{\text{d}}}$-dimensional. If we fix all $N_{\text{d}}$ subsystem symmetries $S_i$ we end up in a $2^{2N_{\text{d}}}$-dimensional Hilbert space as well and can thus find a bijection between these two spaces. The mapping (6) implies

$$
\begin{aligned}
\sigma_{i,A}^z &\rightarrow \tau_{i-1,b}^x \tau_{i,a}^x \\
\sigma_{i,B}^z &\rightarrow \text{eigenvalue}(S_i)\tau_{i-1,b}^x \tau_{i,a}^x \\
\sigma_{i,C}^z &\rightarrow \tau_{i,a}^x \tau_{i,b}^x \, .
\end{aligned}
\tag{7}
$$

Thus all terms are mapped to terms appearing in the transverse-field Ising model, with a two atomic unit cell. This mapping is still somewhat misleading, as the limit $J \rightarrow 0$ would now imply that the model before the mapping has one ground state, while after the mapping it

would have two. This problem can be resolved by realising, that the mapping provided so far is not injective, as it would map the state $|\Phi\rangle$ and $S_{\text{flip}}|\Phi\rangle$ to the same state in the Hilbert space of pseudo-spins. We can now further restrict us to the original Hilbert space in a given symmetry sector of $S_{\text{flip}}$. The mapping between the $2^{2N_d-1}$-dimensional Hilbert space and its image of the same dimension in the twice as large Hilbert space of pseudo-spins is then bijective, linear, and thus conserves the eigenvalues. We therefore have to further adapt the above mapping by replacing $\sigma^z_{N_d,C} \rightarrow \text{eigenvalue}(S_{\text{flip}})\tau^x_{N_d,a}\tau^x_{N_d,b}$ depending of the symmetry sector of $S_{\text{flip}}$.

Notice that the difference between the mapped models in the two symmetry sectors with all local symmetry eigenvalues $s_i$ equal to $+1$ and all local symmetry eigenvalues $s_i$ equal to $+1$ except one is the difference between an open and periodic model after mapping. This follows from the fact that the term $\sigma^z_{i,A}+\sigma^z_{i,B}$ is mapped to $(1+\text{eigenvalue}(S_i))\tau^x_{i-1,b}\tau^x_{i,a}$, which is just zero, if eigenvalue$(S_i)=-1$, thus cutting the periodic chain into an open chain. Similarly, the presence of multiple immobile excitations correspond to setting multiple local symmetries $S_i$ to eigenvalue $s_i=-1$, cutting the transverse-field Ising chain into several disconnected pieces. The mapped Hamiltonian can then be written in terms of pseudo-spins as

$$H_{\text{dual}} = J\sum_{i=1}^{N_d} \tau^z_{i,a} + \tau^z_{i,b} + h\sum_{i=1}^{N_d} \tau^x_{i,a}\tau^x_{i,b} + h\sum_{i=1}^{N_d}(1+s_i)\,\tau^x_{i,b}\tau^x_{i+1,a}\,. \tag{8}$$

## 3  Analytic solution of the low-energy spectrum

In this section we derive an analytic expression for the energy of the immobile excitations discussed. To achieve this, we will calculate the ground-state energy on a finite chain of Hamiltonian (1) in the two relevant symmetry sectors discussed above, take the difference and then take the thermodynamic limit by increasing the system size to infinite.

The ground-state of the ordered phase is contained in the symmetry sector where all symmetries have eigenvalue $+1$. In this symmetry sector the model maps to the pseudo-spin transverse-field Ising chain with alternating Ising coupling strengths and periodic boundary conditions. Its diagonalisation is well known and we will just summarize it shortly in Subsec. 3.1.

The immobile excitation is contained in the symmetry sector where all symmetries have eigenvalue $+1$ except one. In this symmetry sector the model maps to a pseudo-spin transverse-field Ising chain with open boundary conditions. Its approximate but sufficient solution on a finite chain requires more attention and is discussed in detail in Subsec. 3.2.

### 3.1  Solution of periodic case

In the symmetry sector with all eigenvalues $+1$, the pseudo-spin Hamiltonian (8) is given by

$$H = J\sum_{i=1}^{N_d} \tau^z_{i,a} + \tau^z_{i,b} + h\sum_{i=1}^{N_d} \tau^x_{i,a}\tau^x_{i,b} + 2h\sum_{i=1}^{N_d} \tau^x_{i,b}\tau^x_{i+1,a}\,, \tag{9}$$

with $N_d+1=1$. Its diagonalisation can be performed analytically by applying a Jordan-Wigner transformation, Fourier transformation, and Bogoliubov transformation in succession.
The Jordan-Wigner transformation to fermions is performed by ordering the pseudo-spins in the order $(1,a) \rightarrow (1,b) \rightarrow (2,a) \rightarrow (2,b) \rightarrow \dots$ so that one has

$$a^\dagger_{i,a} = e^{i\pi\sum_{j=1}^{i-1}(n_{j,a}+n_{j,b})}c^\dagger_{i,a}$$
$$a^\dagger_{i,b} = e^{i\pi(\sum_{j=1}^{i-1}(n_{j,a}+n_{j,b})+n_{i,a})}c^\dagger_{i,b}\,. \tag{10}$$

The transformed fermionic Hamiltonian reads

$$
\begin{aligned}
H = J \sum_{i=1}^{N_d} (c_{i,a}^\dagger c_{i,a} - c_{i,a} c_{i,a}^\dagger + c_{i,b}^\dagger c_{i,b} - c_{i,b} c_{i,b}^\dagger) \\
+ h \sum_{i=1}^{N_d} (c_{i,a}^\dagger - c_{i,a})(c_{i,b}^\dagger + c_{i,b}) + 2h \sum_{i=1}^{N_d} (c_{i,b}^\dagger - c_{i,b})(c_{i+1,a}^\dagger + c_{i+1,a}) \,.
\end{aligned}
\tag{11}
$$

Note that $S_{\text{flip}}$ maps to $e^{i\pi \sum_{i=1}^{N_d} (n_{j,a} + n_{j,b})}$. If we are in the symmetry sector with $S_{\text{flip}} = +1$, we do not get a negative sign for the Ising interaction where we roll over with the indices. This also shows that physical excitations are those with an even number of fermionic excitations (see also Subsec. 2.4 ).

Next we perform a Fourier transformation

$$
c_{k,a/b}^\dagger = \frac{1}{\sqrt{N_d}} \sum_{j=1}^{N_d} e^{ijk} c_{j,a/b}^\dagger \,,
\tag{12}
$$

where the allowed values of the crystal momentum are $k \in \{\frac{2\pi}{N_d} m ; m \in \{0, 1, ..., N_d - 1\}\}$. The Hamiltonian after the Fourier transformation reads

$$
\begin{aligned}
H = J \sum_k (c_{k,a}^\dagger c_{k,a} - c_{k,a} c_{k,a}^\dagger + c_{k,b}^\dagger c_{k,b} - c_{k,b} c_{k,b}^\dagger) \\
+ h \sum_k c_{k,a}^\dagger c_{-k,b}^\dagger - c_{k,a} c_{-k,b} - c_{k,a} c_{k,b}^\dagger + c_{k,a}^\dagger c_{k,b} \\
+ 2h \sum_k e^{ik} c_{-k,b}^\dagger c_{k,a}^\dagger - e^{-ik} c_{-k,b} c_{k,a} - e^{ik} c_{k,b} c_{k,a}^\dagger + e^{-ik} c_{k,b}^\dagger c_{k,a} \,.
\end{aligned}
\tag{13}
$$

We define the vector

$$
\vec{v}_k = \begin{pmatrix} c_{k,a} & c_{-k,a} & c_{k,b} & c_{-k,b} & c_{k,a}^\dagger & c_{-k,a}^\dagger & c_{k,b}^\dagger & c_{-k,b}^\dagger \end{pmatrix}^{\mathrm{T}}
\tag{14}
$$

to write the Hamiltonian in the form

$$
H = \frac{1}{2} \sum_k \vec{v}_k^\dagger M \vec{v}_k
\tag{15}
$$

with the matrix

$$
M = \begin{pmatrix}
J & 0 & f_+ & 0 & 0 & 0 & 0 & f_- \\
0 & J & 0 & f_+^* & 0 & 0 & f_-^* & 0 \\
f_+^* & 0 & J & 0 & 0 & -f_-^* & 0 & 0 \\
0 & f_+ & 0 & J & -f_- & 0 & 0 & 0 \\
0 & 0 & 0 & -f_-^* & -J & 0 & -f_+^* & 0 \\
0 & 0 & -f_- & 0 & 0 & -J & 0 & -f_+ \\
0 & f_- & 0 & 0 & -f_+ & 0 & -J & 0 \\
f_-^* & 0 & 0 & 0 & 0 & -f_+^* & 0 & -J
\end{pmatrix} ,
\tag{16}
$$

where

$$
f_\pm = h \left( \frac{1}{2} \pm e^{ik} \right)
\tag{17}
$$

The Hamiltonian can then be diagonalized by a fermionic Bogoliubov transformation. One obtains

$$
H = \sum_k 2\epsilon_1(k) \left( \eta_{k,1}^\dagger \eta_{k,1} - \frac{1}{2} \right) + 2\epsilon_2(k) \left( \eta_{k,2}^\dagger \eta_{k,2} - \frac{1}{2} \right)
\tag{18}
$$

where $\epsilon_{2/1}(k)$ are the positive eigenvalues of $M$. These are given by

$$\epsilon_{2/1}(k) = \sqrt{J^2 + \frac{5}{2}h^2 \pm \sqrt{\frac{9}{4}h^4 + (5 + 4\cos(k))(Jh)^2}} \,. \tag{19}$$

Here $\epsilon_1(k)$ corresponds to a single spin-flip excitation moving along the single sites and $\epsilon_2(k)$ corresponds to two spin-flips on a dimer moving along the dimer sites. As further discussed in Sec. 4 the gap $\epsilon_1(k)$ of single spin-flip excitations closes at $(J/h)_c = \sqrt{2}$ corresponding to a second-order phases transition as mention in Subsec. 2.2.

Finally, the ground-state energy is given by

$$E_0^{\text{periodic}}(N_d) = -\sum_{m=0}^{N_d-1} \epsilon_1\left(\frac{2\pi}{N_d}m\right) + \epsilon_2\left(\frac{2\pi}{N_d}m\right) \,. \tag{20}$$

Note that we did not check, that the ground-state defined by the condition $\eta_{k,1/2}\,|0\rangle = 0$ for all crystal momenta $k$ is in fact in the symmetry sector with an even number of fermions as we assumed. For our purposes in this paper, in particular determining the energy of the immobile excitations in the ordered phase, we can argue that the ground-state lies in the symmetry sector with an even number of fermions as does the ground-state in both limiting cases $h/J \to 0$ and $h/J \to \infty$. One further finds that in the thermodynamic limit the difference between the ground-state energy of the two different sectors vanishes and thus plays no role in the energy differences calculated later.

## 3.2 Solution of the open case

Next we consider the mapped model in the symmetry sector with $S_1 = -1$, which according to Eq. (8) is given by

$$H = J\sum_{i=1}^{N_d} \tau_{i,a}^z + \tau_{i,b}^z + h\sum_{i=1}^{N_d} \tau_{i,a}^x\tau_{i,b}^x + 2h\sum_{i=1}^{N_d-1} \tau_{i,b}^x\tau_{i+1,a}^x \,. \tag{21}$$

We can again perform the Jordan-Wigner transformation introduced in the previous subsection to obtain the Hamiltonian

$$\begin{aligned}
H = {} & J\sum_{i=1}^{N_d}(c_{i,a}^\dagger c_{i,a} - c_{i,a}c_{i,a}^\dagger + c_{i,b}^\dagger c_{i,b} - c_{i,b}c_{i,b}^\dagger) \\
& + h\sum_{i=1}^{N_d}(c_{i,a}^\dagger - c_{i,a})(c_{i,b}^\dagger + c_{i,b}) + 2h\sum_{i=1}^{N_d-1}(c_{i,b}^\dagger - c_{i,b})(c_{i+1,a}^\dagger + c_{i+1,a}) \,.
\end{aligned} \tag{22}$$

In the open-chain case the dual pseudo-spin model is the same for $S_{\text{flip}} = \pm 1$ and therefore also the ground-state energy is the same in both symmetry sectors.

To diagonalize the pseudo-spin model in this case we perform a real space Bogoliubov transformation following [20]. First, we introduce the vector

$$\Psi_i = \begin{pmatrix} c_{i,a} \\ c_{i,b} \end{pmatrix} \tag{23}$$

and write the Hamiltonian in the form

$$H = \sum_{i,j=1}^{N_d} \Psi_i^\dagger A_{ij}\Psi_j - \Psi_i^T A_{ij}\left(\Psi_j^\dagger\right)^T + \Psi_i^\dagger B_{ij}\left(\Psi_j^\dagger\right)^T + \Psi_i^T B_{ij}\Psi_j \,, \tag{24}$$

where $A_{ij}$ and $B_{ij}$ are two by two matrices given by

$$
\begin{aligned}
A_{ij} &= \delta_{i,j} \frac{1}{2} \begin{pmatrix} 2J & h \\ h & 2J \end{pmatrix} + \delta_{i+1,j} \frac{1}{2} \begin{pmatrix} 0 & 0 \\ 2h & 0 \end{pmatrix} + \delta_{i,j+1} \frac{1}{2} \begin{pmatrix} 0 & 2h \\ 0 & 0 \end{pmatrix} \\
B_{ij} &= \delta_{i,j} \frac{1}{2} \begin{pmatrix} 0 & h \\ -h & 0 \end{pmatrix} + \delta_{i+1,j} \frac{1}{2} \begin{pmatrix} 0 & 0 \\ 2h & 0 \end{pmatrix} + \delta_{i,j+1} \frac{1}{2} \begin{pmatrix} 0 & -2h \\ 0 & 0 \end{pmatrix} .
\end{aligned}
\tag{25}
$$

Next we introduce new fermionic operators given by

$$
\eta_j = \sum_{i=1}^{N_{\mathrm{d}}} g_{ji} \Psi_i + m_{ji} \left( \Psi_i^\dagger \right)^T ,
\tag{26}
$$

where $g_{ji}, m_{ji}$ are again two by two matrices. We then write the Hamiltonian in the form

$$
H = \sum_j \eta_j^\dagger \Gamma_j \eta_j + E_0^{\mathrm{open}} .
\tag{27}
$$

Because $H$ turns into $-H$ under particles-hole exchange $c^\dagger \leftrightarrow c$, the spectrum of $H$ is symmetric around zero and the ground-state energy $E_0^{\mathrm{open}}$ is just given by $-\sum_j \frac{1}{2} \mathrm{Tr}(\Gamma_j)$. We now calculate the commutator $[H, \eta_j]$ using this new form which yields

$$
[H, \eta_j] = -\Gamma_j \eta_j .
\tag{28}
$$

Next we insert the definition of $\eta_j$ into both sides of this equation to obtain the coupled equations

$$
\begin{aligned}
\Gamma X &= Y 2(A+B) \\
\Gamma Y &= X 2(A-B) ,
\end{aligned}
\tag{29}
$$

where $X = g + m$ and $Y = g - m$. The quantities without indices are the matrices with two-by-two matrices as elements. As these two equations determine each other, we can calculate $Y$ via

$$
Y 2(A+B) 2(A-B) = \Gamma^2 Y .
\tag{30}
$$

This is just the eigenvalue equation written in terms of a matrix (here $Y$) containing the eigenvectors. We will thus determine the eigenvalues of $(A+B)(A-B)$ to obtain the eigenvalues of the initial Hamiltonian. Writing out this matrix we obtain

$$
(A+B)(A-B) = \begin{pmatrix}
J^2 + h^2 & Jh & 0 & 0 & 0 & 0 & \dots & 0 \\
Jh & J^2 + 4h^2 & 2Jh & 0 & 0 & 0 & \dots & 0 \\
0 & 2Jh & J^2 + h^2 & Jh & 0 & 0 & \dots & 0 \\
0 & 0 & Jh & J^2 + 4h^2 & 2Jh & 0 & \dots & 0 \\
0 & 0 & 0 & 2Jh & \ddots & \ddots & \dots & 0 \\
0 & 0 & 0 & 0 & \ddots & \ddots & 2Jh & 0 \\
0 & 0 & 0 & 0 & \dots & 2Jh & J^2 + h^2 & Jh \\
0 & 0 & 0 & 0 & 0 & \dots & Jh & J^2
\end{pmatrix} .
\tag{31}
$$

This is almost a tridiagonal matrix with alternating entries on the three diagonals with an exception in the last entry: Here one has $J^2$ instead of $J^2 + 4h^2$. The size of this matrix is $2N_{\mathrm{d}}$ by $2N_{\mathrm{d}}$. We define the determinant of this $2N_{\mathrm{d}}$ by $2N_{\mathrm{d}}$ matrix minus $\lambda^2$ as

$$
P_{N_{\mathrm{d}}} = \det \left( (A+B)(A-B)_{2N_{\mathrm{d}} x 2N_{\mathrm{d}}} - \lambda^2 \mathbf{1}_{2N_{\mathrm{d}} x 2N_{\mathrm{d}}} \right) .
\tag{32}
$$

The analogous determinant for a $(2N_\mathrm{d}-1)$ by $(2N_\mathrm{d}-1)$ matrix, which is given by the above matrix with the first row and column removed, thus starting with the entry $J^2 + 4h^2$ in the top left, will be called $Q_{N_\mathrm{d}}$. By repeated Laplace expansion in its first row and column, we find the recursion relations

$$
\begin{aligned}
P_{N_\mathrm{d}} &= (J^2 + h^2 - \lambda^2)Q_{N_\mathrm{d}} - (Jh)^2 P_{N_\mathrm{d}-1} \\
Q_{N_\mathrm{d}} &= (J^2 + 4h^2 - \lambda^2)P_{N_\mathrm{d}-1} - (2Jh)^2 Q_{N_\mathrm{d}-1} \,.
\end{aligned}
\tag{33}
$$

Inserting the second into the first equation and defining the vector

$$
\vec{w}_{N_\mathrm{d}} = \begin{pmatrix} P_{N_\mathrm{d}} \\ Q_{N_\mathrm{d}} \end{pmatrix} ,
\tag{34}
$$

we rewrite this equation as

$$
\begin{aligned}
\vec{w}_{N_\mathrm{d}} &= M \vec{w}_{N_\mathrm{d}-1} \quad \text{with} \\
\vec{w}_0 &= \begin{pmatrix} 1 \\ \frac{1}{J^2} \end{pmatrix} \\
M &= \begin{pmatrix} (J^2 + 4h^2 - \lambda^2)(J^2 + h^2 - \lambda^2) - (Jh)^2 & -(J^2 + h^2 - \lambda^2)(2Jh)^2 \\ (J^2 + 4h^2 - \lambda^2) & -(2Jh)^2 \end{pmatrix} .
\end{aligned}
\tag{35}
$$

This recursion equation can be solved by diagonalising $M$ which yields

$$
\begin{aligned}
P_{N_\mathrm{d}} &= \frac{1}{\omega_2 - \omega_1}\left( \omega_2^{N_\mathrm{d}+1} - \omega_1^{N_\mathrm{d}+1} + 4h^2\left(\lambda^2 - h^2\right)\left(\omega_2^{N_\mathrm{d}} - \omega_1^{N_\mathrm{d}}\right)\right) \\
\omega_{1/2} &= \frac{\mathrm{Tr}(M)}{2} \pm \sqrt{\left(\frac{\mathrm{Tr}(M)}{2}\right)^2 - \det(M)} \\
\mathrm{Tr}(M) &= J^4 + 4h^4 - \left(2J^2 + 5h^2\right)\lambda^2 + \lambda^4 \\
\det(M) &= 4\left(Jh\right)^4 ,
\end{aligned}
\tag{36}
$$

where $\omega_{1/2}$ are the eigenvalues of $M$. We find the eigenvalues via the equation $P_{N_\mathrm{d}} = 0$. To solve this equation we consider two different cases. First, we look at the case $|\omega_1| \neq |\omega_2|$ where we find

$$
0 = \max(\omega_1, \omega_2) + 4h^2\left(\lambda^2 - h^2\right)
\tag{37}
$$

in the limit $N_\mathrm{d} \to \infty$. This equation has two roots. A single root at $\lambda = 0$ and a quadratic root at $\lambda = \sqrt{J^2 + h^2}$. Second, we consider the case $|\omega_1| = |\omega_2|$. This is only possible, if both roots are complex and we can set

$$
\omega_{1/2} = |\omega_{1/2}|e^{\pm i\phi} = 2(Jh)^2 e^{\pm i\phi} \,.
\tag{38}
$$

The eigenvalue equation can then be rewritten as

$$
\frac{\sin((N_\mathrm{d}+1)\phi)}{\sin(N_\mathrm{d}\phi)} = -2\left(1 + \frac{3}{2}\left(\frac{h}{J}\right)^2 \pm \sqrt{\frac{9}{4}\left(\frac{h}{J}\right)^4 + (5 + 4\cos(\phi))\left(\frac{h}{J}\right)^2}\right)
\tag{39}
$$

in terms of the angle $\phi$. To first order, we can approximate the function on the left in the thermodynamic limit as a sequence of vertical lines at the roots of $\sin((N_\mathrm{d}+1)\phi)$. Exceptions are the first positive root and the last root before $\phi = \pi$, which are vertical lines that do not extend infinitely but start (end) at $+1$ ($-1$).

Let us focus on the case $|h| > \frac{1}{\sqrt{2}}|J|$ where the gap closes. Here, the negative mode (negative

sign in Eq. (39)) does not include the first root. In addition, both modes always miss the last root. Thus, we obtain the roots

$$\epsilon_1\left(\frac{\pi}{N_\mathrm{d}+1}m\right) \text{ for } m \in \{2,...,N_\mathrm{d}-1\}$$
$$\epsilon_2\left(\frac{\pi}{N_\mathrm{d}+1}m\right) \text{ for } m \in \{1,...,N_\mathrm{d}-1\} \,. \tag{40}$$

This turns out to be insufficient and we will need the corrections to these roots in order $1/(N_\mathrm{d}+1)$, which will contribute an additional finite term in the energy of the immobile excitation in the thermodynamic limit. To do so, we define the roots $\phi_m = \pi m/(N_\mathrm{d}+1)$ and make the ansatz

$$\phi = \phi_m + c_{2/1}(\phi_m)\frac{1}{N_\mathrm{d}+1} + O\left(\frac{1}{(N_\mathrm{d}+1)^2}\right) \,, \tag{41}$$

as an expansion around the root $\phi_m$. Note that $c_{2/1}$ is a function of $\phi_m$. Inserting this in Eq. (39), expanding in powers of $1/(N_\mathrm{d}+1)$ and comparing the zeroth order determines $c_{2/1}$ via the equation

$$f_{2/1}(\phi) = -2\left(1 + \frac{3}{2}\left(\frac{h}{J}\right)^2 \pm \sqrt{\frac{9}{4}\left(\frac{h}{J}\right)^4 + (5+4\cos(\phi))\left(\frac{h}{J}\right)^2}\right)$$
$$\tan(c_{2/1}(\phi_m)) = \frac{\sin(\phi_m)}{\cos(\phi_m) - \frac{1}{f_{2/1}(\phi_m)}} \,. \tag{42}$$

The functions $c_{2/1}(\phi_m)$ are then fully determined by the requirements $|c_1| < \pi$ and

$$\mathrm{sign}(c_{2/1}(\phi_m)) = -\mathrm{sign}(f_{2/1}(\phi_m)).$$

The ground-state energy $E_0^\mathrm{open}$ is thus given by

$$E_0^\mathrm{open}(N_\mathrm{d}) = -2\sqrt{J^2+h^2} - \sum_{m=2}^{N_\mathrm{d}-1} \epsilon_1\left(\phi_m + c_1(\phi_m)\frac{1}{N_\mathrm{d}+1}\right)$$
$$-\sum_{m=1}^{N_\mathrm{d}-1} \epsilon_2\left(\phi_m + c_2(\phi_m)\frac{1}{N_\mathrm{d}+1}\right) + O\left(\frac{1}{N_\mathrm{d}+1}\right)$$
$$= -2\sqrt{J^2+h^2} - \sum_{m=2}^{N_\mathrm{d}-1} \epsilon_1(\phi_m) - \sum_{m=1}^{N_\mathrm{d}-1} \epsilon_2(\phi_m)$$
$$-\frac{1}{N_\mathrm{d}+1}\sum_{m=2}^{N_\mathrm{d}-1} \epsilon_1'(\phi_m)c_1(\phi_m) - \frac{1}{N_\mathrm{d}+1}\sum_{m=1}^{N_\mathrm{d}-1} \epsilon_2'(\phi_m)c_2(\phi_m) + O\left(\frac{1}{N_\mathrm{d}+1}\right) \,. \tag{43}$$

### 3.3 Energy of the immobile mode

We calculate the energy of the immobile mode via

$$\epsilon_\mathrm{immobile} = \lim_{N_\mathrm{d}\to\infty} E_0^\mathrm{open}(N_\mathrm{d}) - E_0^\mathrm{periodic}(N_\mathrm{d}) \,. \tag{44}$$

First, we consider the thermodynamic limit of the following sums contained in the ground-state energy $E_0^{\text{open}}$ of the open chain given by Eq. (43). Applying partial integration gives

$$
\begin{aligned}
&\lim_{N_{\text{d}} \to \infty} -\frac{1}{N_{\text{d}}+1} \sum_{m=2}^{N_{\text{d}}-1} \epsilon_1'(\phi_m) c_1(\phi_m) - \frac{1}{N_{\text{d}}+1} \sum_{m=1}^{N_{\text{d}}-1} \epsilon_2'(\phi_m) c_2(\phi_m) \\
&= \lim_{N_{\text{d}} \to \infty} -\sum_{n \in \{1,2\}} \frac{1}{N_{\text{d}}+1} \sum_{m=0}^{N_{\text{d}}} \epsilon_n'(\phi_m) c_n(\phi_m) \\
&= -\sum_{n=1}^{2} \frac{1}{\pi} \int_0^{\pi} \epsilon_n'(\phi) c_n(\phi) d\phi \\
&= -\sum_{n=1}^{2} \frac{1}{\pi} [\epsilon_n(\phi) c_n(\phi)]_0^{\pi} - \frac{1}{\pi} \int_0^{\pi} \epsilon_n(\phi) c_n'(\phi) d\phi \\
&= -\epsilon_1(0) - \epsilon_1(\pi) - \epsilon_2(\pi) + \sum_{n=1}^{2} \frac{1}{\pi} \int_0^{\pi} \epsilon_n(\phi) c_n'(\phi) d\phi \ ,
\end{aligned}
\tag{45}
$$

where we used that above the quantum critical point at $(J/h)_{\text{c}}$ we have $c_1(0) = -\pi$, $c_1(\pi) = \pi$, $4c_2(0) = 0$, and $c_2(\pi) = \pi$, according to the appropriate values of the inverse tangent in Eq. (42). Next we can subtract the exothermic contributions of the ground-state energies for open and closed boundary conditions by using the symmetry about $k = \pi/2$ and assuming $N_{\text{d}}$ to be even to obtain

$$
\begin{aligned}
\epsilon_{\text{immobile}} = \ &-2\sqrt{J^2 + h^2} + \epsilon_1(0) + \epsilon_1(\pi) + \epsilon_2(\pi) \\
&+ \lim_{N_{\text{d}} \to \infty} \sum_{n=1}^{2} \sum_{m=1}^{N_{\text{d}}/2-1} 2\epsilon_n(\frac{\pi}{N_{\text{d}}} 2m) - \epsilon_n(\frac{\pi}{N_{\text{d}}+1} 2m) - \epsilon_n(\frac{\pi}{N_{\text{d}}+1}(2m+1)) \\
&- \epsilon_1(0) - \epsilon_1(\pi) - \epsilon_2(\pi) + \sum_{n \in \{1,2\}} \frac{1}{\pi} \int_0^{\pi} \epsilon_n(\phi) c_n'(\phi) d\phi \ .
\end{aligned}
\tag{46}
$$

Finally, a Taylor expansion of the last two terms in the sum about $2\pi/N_{\text{d}}$ yields in the limit $N_{\text{d}} \to \infty$

$$
\epsilon_{\text{immobile}} = -2\sqrt{J^2 + h^2} + \sum_{n=1}^{2} \left( \frac{1}{2}(\epsilon_n(\pi) + \epsilon_n(0)) - \frac{1}{\pi} \int_0^{\pi} \epsilon_n(x)\left(1 - c_n'(\phi)\right) dx \right) .
\tag{47}
$$

With $\epsilon_n$ with $n \in \{1,2\}$ being the expressions for the two mobile modes given by Eq. (19). Note that $\epsilon_1$ is the lower lying mode. We confirmed this result by comparison with the series expansion about the high-field limit up to sixth order.

## 4 Phase transition and excitation spectra

The low-energy spectrum contains three distinct modes. Two mobile modes, which in the limit of large fields $h \gg J$ are adiabatically connected to a single spin-flip located on a single site or two spin-flips on a dimer, respectively. The gap of the single spin-flip excitation closes at $J/h = \sqrt{2}$ resulting in a second-order phase transition. Further, one immobile mode energetically located between these two modes that could overlap at certain crystal moments (see Fig. 2). Both, the excitation gap of the low-lying mobile mode and the immobile mode decrease in energy when approaching the quantum critical point as shown in Fig. 3. As already

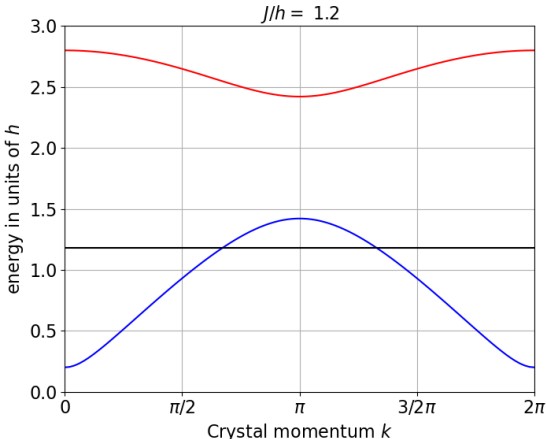

Figure 2: Dispersion of the three modes of the model at $J/h = 1.2$. The upper mode (red line) is adiabatically connected to two spin-flips on a dimer, the lower mode (blue line) is adiabatically connected to a single spin-flip on a single site. The immobile mode (black line) has a flat dispersion and lies energetically between the two mobile modes but can overlap with the two mobile modes.

stressed above, the immobile modes are protected by the local symmetries and correspond to eigenvalues $s_i = -1$ (see Subsec. 2.1). A closing of the gap between the energy of such immobile modes and the ground-state would imply a phase with a broken local symmetry, which is forbidden by Elitzur's theorem [19]. Our findings for Eq. (1) and variants of it discussed in Appendix B are therefore fully consistent in this respect. Consequently, to detect a gap closing of a mode exhibiting dimensional reduction, one has to investigate systems in higher spatial dimensions.

The duality mapping to the transverse-field Ising model Eq. (2.4) in terms of pseudo-spins

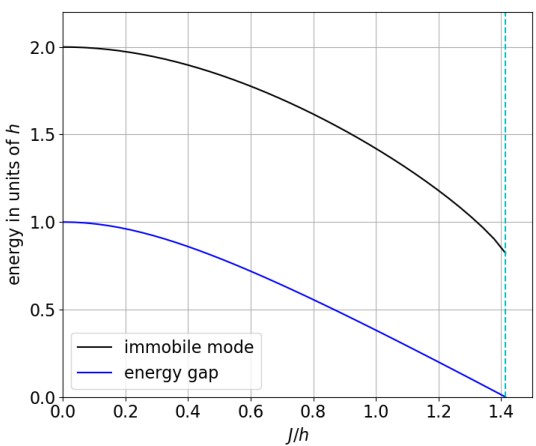

Figure 3: Energy gaps of the lower lying mobile mode (blue line) and the immobile mode (black line) as a function of the parameter $J/h$.

yields an interpretation of our immobile modes as walls between different Ising chain segments. So far we looked at a single immobile mode. Considering two immobile excitations,

the energy of a given configuration depends on the distance between these two immobile modes or the distance of the two walls in the dual pseudo-spin formulation. Following the previous derivation, we find that the energy of such a two-mode configuration is given as the difference between the ground-state energy of the periodic transverse field Ising model with $N_d$ dimers and the sum of the ground-state energies of two open transverse-field Ising chain segments with $d$ and $N_d - d$ dimers, respectively. Taking this difference and considering the thermodynamic limit, we find a monotonically decreasing dependence of the energy on the distance $d$ between them. In this wall picture one might interpret this as a force similar to a Casimir force between conducting plates, while in the original model the interpretation would be an attractive interaction induced via the three-spin interaction. The force is defined as the energy of two immobile modes at a distance $d$ minus the energy of these two modes at a distance $d - 1$. These forces are given in the thermodynamic limit by the expression

$$F_\infty(d) = E_0^{\text{open}}(d - 1) - E_0^{\text{open}}(d) + \sum_{n=1}^{2} \frac{1}{\pi} \int_0^\pi \epsilon_n(k) dk , \tag{48}$$

where $E_0^{\text{open}}(d)$ is the ground-state energy of the transverse-field Ising chain with open boundary conditions (43) with $d$ dimers. A derivation is given in Appendix A. The ground-state energies of the finite open chain segments are calculated numerically and the resulting forces are illustrated in Fig. 4. One notices the dependence on the parameter $J/h$. While smaller $J/h$ leads to stronger forces at small distances that quickly decay, larger values lead to smaller forces with slower decay.

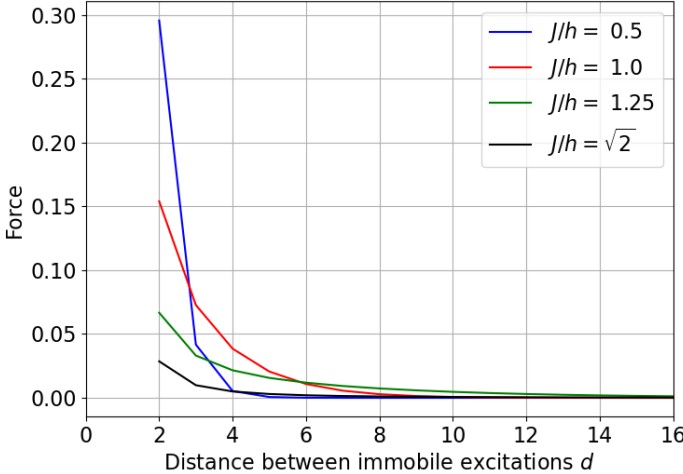

Figure 4: Force between two immobile excitations separated by a distance of $d$ dimers for different values of $J/h$. Due to the discrete nature of the model the force is calculated as the difference between the energy of two immobile excitations at a distance $d$ and $d - 1$.

## 5  Conclusions

In this work we introduced a quantum spin model with three-spin interactions on a diamond chain. The model is analytically solvable due to the existence of an extensive number of local

symmetries allowing an exact duality mapping to pseudo-spins 1/2 transverse-field Ising chain segments. The ground-state phase diagram displays a second-order phase transition in the 2D Ising universality class which is driven by mobile excitations of the system. Interestingly, fully immobile excitations exist which are protected by the local symmetries. In the dual language these immobile excitations correspond to vacancies so that each immobile excitations cuts the dual transverse-field Ising chain. Further, we have obtained analytic expressions for the attractive Casimir-like force between two immobile excitations.

We find particular interesting the exact fragmentation in terms of open chain segments in the dual formulation for an arbitrary number of immobile excitations. This paves the way towards the investigation of finite density and temperature properties by building on the analytic findings derived in this work. In addition, the derived duality links dilution disorder in the transverse field Ising chain with disorder in the placement of immobile excitations in the original model, which is an interesting topic for future investigations.

In the future we plan to investigate extensions of our model to higher dimensions. Indeed, in two dimensions one obtains a model of three-spin interactions on the 2D pyrochlore lattice which is isospectral to the XY toric code (XYTC) in a transverse magnetic field [16]. The XYTC features fractonic excitations which are protected by subsystem symmetries and it is therefore interesting to understand their fate upon an increasing external magnetic field.

## Acknowledgements

We acknowledge fruitful exchange with Konstantinos Sfairopoulos.

**Funding information** KPS acknowledges the support by the Munich Quantum Valley, which is supported by the Bavarian state government with funds from the Hightech Agenda Bayern Plus.

**Data availability** Supplementary data for all figures are available online.

## A  Force between two immobile excitations

In order to calculate the force between two (or more) immobile excitations we calculate the energy of a configuration of two immobile modes the same way we did before by subtracting the ground-state energy of the sector without immobile excitations from the ground-state energy of the sector with two modes at a distance $d$ from each other. In the sector with two immobile excitations the dual pseudo-spin model (see 2.3) becomes an open transverse-field Ising chain segment with $d$ dimers and another open chain segment with $N - d$ dimers. Thus, the energy of the configuration is given by

$$E_2(d) = \lim_{N \to \infty} \left[ E_0^{\text{open}}(N - d) + E_0^{\text{open}}(d) - E_0^{\text{periodic}}(N) \right] . \tag{49}$$

We define the force between two immobile excitations as the difference of the energies of a configuration with two immobile excitations at a distance $d$ and one with distance $d - 1$

$$
\begin{aligned}
F_\infty(d) &= -\left[ E_2(d) - E_2(d-1) \right] \\
&= \lim_{N \to \infty} \left[ E_0^{\text{open}}(N - d + 1) + E_0^{\text{open}}(d-1) - E_0^{\text{open}}(N - d) - E_0^{\text{open}}(d) \right] \\
&= E_0^{\text{open}}(d-1) - E_0^{\text{open}}(d) + \lim_{N \to \infty} \left[ E_0^{\text{open}}(N - d + 1) - E_0^{\text{open}}(N - d) \right] .
\end{aligned}
\tag{50}
$$

Considering expression Eq. 43 we can calculate the limit in this expression exactly. To do so we use the following identity for an analytic function $f$ on the interval $(0,1)$

$$\lim_{N\to\infty} \sum_{i=1}^{N} f\left(\frac{i}{N}\right) - \sum_{i=1}^{N-d} f\left(\frac{i}{N-d}\right) = d\int_0^1 f(x)dx \ . \tag{51}$$

A prove of this statement is simply obtained by Taylor expanding the functions and using

$$\sum_{i=1}^{N} i^k = \frac{1}{k+1}N^{k+1} + \frac{1}{2}N^k + O(N^{k-1}) \ . \tag{52}$$

## B   Alternating magnetic field strengths

In this part we generalize our model to different magnetic fields $h_1$ and $h_2$ on the single and on the dimer sites. We call the magnetic field on the single sites $h_1 = h$ and the magnetic field on the dimers $h_2 = rh$ with the ratio $r$ of these magnetic fields. The calculations are mostly analogous to the main body of the article. Here we will summarize and expand on other aspects not mentioned previously.

### B.1   Periodic boundary conditions

Here the calculations are fully analogous we just replace the functions $f_+$ and $f_-$ with

$$\begin{aligned} f_+ &= h\left(\frac{1}{2} + re^{ik}\right) \\ f_- &= h\left(\frac{1}{2} - re^{ik}\right) \ . \end{aligned} \tag{53}$$

In general the eigenvalues of the matrix $M$ in Eq. 16 are given by

$$\begin{aligned} \epsilon_{2/1} &= \sqrt{J^2 + |f_+|^2 + |f_-|^2 \pm \sqrt{4J^2|f_+|^2 + 2|f_+|^2|f_-|^2 + 2\Re\left(\left(f_+^* f_-\right)^2\right)}} \\ &= \sqrt{J^2 + \left(\frac{1}{2} + 2r^2\right)h^2 \pm \sqrt{\left(2r^2 - \frac{1}{2}\right)^2 h^4 + (1 + 4r^2 + 4r\cos(k))J^2 h^2}} \end{aligned} \tag{54}$$

with the same allowed values for $k$ as in the previous case $k \in \{\frac{2\pi}{N_d}m; m \in \{0, 1, ..., N_d - 1\}\}$.

### B.2   Open boundary conditions

For this case we simply replace the matrices $A$ and $B$ by

$$\begin{aligned} A_{ij} &= \delta_{i,j}\frac{1}{2}\begin{pmatrix} 2J & h \\ h & 2J \end{pmatrix} + \delta_{i+1,j}\frac{1}{2}\begin{pmatrix} 0 & 0 \\ 2rh & 0 \end{pmatrix} + \delta_{i,j+1}\frac{1}{2}\begin{pmatrix} 0 & 2rh \\ 0 & 0 \end{pmatrix} \\ B_{ij} &= \delta_{i,j}\frac{1}{2}\begin{pmatrix} 0 & h \\ -h & 0 \end{pmatrix} + \delta_{i+1,j}\frac{1}{2}\begin{pmatrix} 0 & 0 \\ 2rh & 0 \end{pmatrix} + \delta_{i,j+1}\frac{1}{2}\begin{pmatrix} 0 & -2rh \\ 0 & 0 \end{pmatrix} \ . \end{aligned} \tag{55}$$

We proceed analogously with the real space Bogoliubov transformation and obtain the recursion relations

$$
\vec{w}_{N_\mathrm{d}} = M \vec{w}_{N_\mathrm{d}-1}
$$
$$
\vec{w}_0 = \begin{pmatrix} 1 \\ \frac{1}{J^2} \end{pmatrix}
$$
$$
M = \begin{pmatrix} (J^2 + 4r^2h^2 - \lambda^2)(J^2 + h^2 - \lambda^2) - (Jh)^2 & -(J^2 + h^2 - \lambda^2)(2rJh)^2 \\ (J^2 + 4r^2h^2 - \lambda^2) & -(2rJh)^2 \end{pmatrix} .
\tag{56}
$$

In general we can write the solution for $P_{N_\mathrm{d}}$ as

$$
\begin{aligned}
P_{N_\mathrm{d}} &= \frac{1}{\omega_2 - \omega_1} \left( P_0 \left( \omega_2^{N_\mathrm{d}+1} - \omega_1^{N_\mathrm{d}+1} \right) + (M_{12}Q_0 + M_{11}P_0 - \mathrm{Tr}(M)P_0) \left( \omega_2^{N_\mathrm{d}} - \omega_1^{N_\mathrm{d}} \right) \right) \\
&= \frac{1}{\omega_2 - \omega_1} \left( \left( \omega_2^{N_\mathrm{d}+1} - \omega_1^{N_\mathrm{d}+1} \right) + 4r^2h^2 \left( \lambda^2 - h^2 \right) \left( \omega_2^{N_\mathrm{d}} - \omega_1^{N_\mathrm{d}} \right) \right) .
\end{aligned}
\tag{57}
$$

We can again calculate the stationary modes via

$$
\max(\omega_1, \omega_2) + 4r^2h^2 \left( \lambda^2 - h^2 \right) = 0
\tag{58}
$$

This has again the roots at $\lambda = 0$ and a quadratic root at $\lambda = \sqrt{J^2 + h^2}$ but this root only exists for $|r| > 1/2$. Next we again consider the case where $|\omega_1| = |\omega_2|$ and we write

$$
\omega_{1/2} = 2|r|J^2h^2 e^{\pm i\phi} ,
\tag{59}
$$

to obtain the analogous equation

$$
\begin{aligned}
&\frac{\sin\left( (N_\mathrm{d} + 1)\phi \right)}{\sin(N_\mathrm{d}\phi)} \\
&= -2r \left( 1 + \left( 2r^2 - \frac{1}{2} \right) \left( \frac{h}{J} \right)^2 \pm \sqrt{ \left( 2r^2 - \frac{1}{2} \right)^2 \left( \frac{h}{J} \right)^4 + (1 + 4r^2 + 4r\cos(\phi)) \left( \frac{h}{J} \right)^2 } \right) .
\end{aligned}
\tag{60}
$$

We analyse the right-hand side for $\phi = 0$ and $\phi = \pi$ to see, if the modes closest to these values are included. We assume $|r| > 1/2$. We find that the positive branch (+ at the $\pm$) is always smaller than $-1$ and thus always misses both crossings. The negative branch again always misses the crossing at $\phi = \pi$ and crosses at $\phi = 0$ for

$$
\left( \frac{h}{J} \right)^2 = \frac{1}{2r} ,
\tag{61}
$$

where we further assumed $r$ to be positive. This is again the point of the quantum phase transition where the mobile mode condenses.

## B.3   The case r < 1/2

Here we remove the double root at $\sqrt{J^2 + h^2}$. We gain an addend of $\epsilon_1(\pi) + \epsilon_2(\pi)$ by now including the last root of Eq. 39 but this addend is again removed due to the changed start and end values of the function

$$
\begin{aligned}
c_1(0) &= -\pi \\
c_1(\pi) &= 0 \\
c_2(0) &= 0 \\
c_2(\pi) &= 0 .
\end{aligned}
\tag{62}
$$

So in summary for $r > 0$ we can write

$$
\epsilon_{\mathrm{immobile}} = -2\Theta(r - 1/2)\sqrt{J^2 + h^2} + \sum_{n=1}^{2} \frac{1}{2} \left( \epsilon_n(\pi) + \epsilon_n(0) \right) - \frac{1}{\pi} \int_0^{\pi} \epsilon_n(x) \left( 1 - c_n'(\phi) \right) dx .
\tag{63}
$$

# C  Second-order degenerate perturbation theory

We discuss the second-order perturbative corrections in $h/J$ to the ground-state sector at $h = 0$. We look at a system with $N_d$ dimers and periodic coupling. The ground-state manifold can be constructed by providing the eigenvalues $\{s_i\}_{i \leq N_d}$ ($s_i \in \{-1, +1\}$) for the $N_d$ local symmetries $S_i$ and the global symmetry $S_{\text{flip}}$. We may construct these states by applying the operators $(\mathbb{1} + s_i S_i)$ to the in x-direction polarised state $|\rightarrow\rangle$ depending on the sign of the eigenvalue of the symmetry $S_i$. We can proceed equivalently with the global symmetry $S_{\text{flip}}$ to obtain a basis of the $2^{N_d+1}$ dimensional ground-state vector space

$$|\Psi_\pm \left(\{s_i\}_{i \leq N_d}\right)\rangle = \left(\mathbb{1} \pm S_{\text{flip}}\right)\prod_{i=1}^{N_d}(\mathbb{1} + s_i S_i)|\rightarrow\rangle \ . \tag{64}$$

We split our Hamiltonian 1 into an unperturbed part $H_0$, which contains the three spin interactions and a perturbation equal to the magnetic field $V = h\sum_{i=1}^{N_d}\sum_{f \in \{A,B,C\}} \sigma_{i,f}^z$. Using Takahashi perturbation theory [21] we can determine the effective Hamiltonian in the degenerate ground-space Hilbert-space up to second order via

$$H^{\text{eff}} = J\left(\frac{1}{J}H_0 + (h/J)P_0 V P_0 + (h/J)^2 P_0 V \frac{1-P_0}{E_0 - H_0} V P_0 + O\left((h/J)^3\right)\right) , \tag{65}$$

where $P_0$ is the projection operator onto the degenerate ground-state Hilbert-space. We note that acting with a single $\sigma^z$ operator on any of the ground-states 64 will flip the eigenvalue of the two three-spin interactions touched by the site the $\sigma^z$ operator acts on and thus this state will no longer be in the degenerate ground-state Hilbert-space. The first order contribution therefore is identical to zero. The second order contribution contains non-vanishing contributions. Considering a dimer and its neighbouring single site we find five ways to subsequently arrange and apply two $\sigma^z$ operators while staying inside the degenerate ground-state Hilbert-space. There are three ways to apply the $\sigma^z$ operators on the same site and two ways to place them on the Dimer (the first on the upper site and the second on the lower site and vice versa). The operator $\frac{1-P_0}{E_0-H_0}$ will always be equal to $-\frac{1}{4}\mathbb{1}$ in this situation, thus considering the five ways to arrange the $\sigma^z$ operators we find

$$H^{\text{eff}} = J\left(\frac{1}{J}H_0 - \frac{1}{4}(h/J)^2 \sum_{i=1}^{N_d}(3\mathbb{1} + 2S_i) + O\left((h/J)^3\right)\right) . \tag{66}$$

Considering

$$-\frac{1}{4}(3\mathbb{1} + 2S_i)(\mathbb{1} + s_i S_i) = \begin{cases} -\frac{5}{4}(\mathbb{1} + s_i S_i) & s_i = +1 \\ -\frac{1}{4}(\mathbb{1} + s_i S_i) & s_i = -1 \end{cases} \tag{67}$$

we find that the $2^{N_d+1}$-fold ground-state degeneracy is in second order reduced to a two-fold degeneracy only governed by the eigenvalue of the global symmetry $S_{\text{flip}}$ given by the two states

$$|\Psi_\pm\rangle = \left(\mathbb{1} \pm S_{\text{flip}}\right)\prod_{i=1}^{N_d}(\mathbb{1} + S_{2i})|\rightarrow\rangle \ . \tag{68}$$

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
