# Peer review of "Immobile and mobile excitations of three-spin interactions on the diamond chain"

_SciPost Physics Core_

## Round 2 · Referee Report · Jozef Strecka (Referee 1) · 2025-12-23

Strengths

1- The extensive local symmetries allow an elegant mapping to fragmented transverse-field Ising chains, which subsequently allows a clear analytic analysis of the nature of the immobile excitations.
2- The derivation of the immobile excitation energy is technically highly nontrivial and constitutes a novel and valuable result.
3- The introduced one-dimensional quantum spin chain captures key ideas of symmetry-protected immobility and Hilbert-space fragmentation that are central to fracton-inspired physics.

Weaknesses

The manuscript does not have any major weakness. Minor weaknesses and a few suggestion atimed at improving the manuscript are mentioned in the part requested changes.

Report

The authors introduce an exactly solvable spin-1/2 model on a diamond chain with three-spin interaction in a transverse magnetic field. The model exhibits both dispersive mobile excitations, which drive a continuous Ising quantum phase transition, and symmetry-protected fully immobile excitations. Using an extensive set of local Z2-symmetries, the Hamiltonian is mapped onto a set of disconnected transverse-field Ising chains. This mapping enables analytic calculation of the excitation spectrum, the phase transition, and Casimir-like interactions between immobile modes. The manuscript provides an interesting example of Hilbert-space fragmentation and symmetry-protected dimensional reduction in a one-dimensional quantum spin chain motivated by fracton physics. The manuscript provides original, technically solid, and conceptually interesting results meeting all acceptance criteria of he SciPost Physics Core journal. With the clarifications and minor improvements suggested in the part requested changes, it will make an excellent contribution to the SciPost Physics Core journal.

Requested changes

1- Although the manuscript is motivated by fracton physics, the immobile excitations do not exhibit some defining properties of fractons. It would improve clarity of the manuscript if the authors explicitly state in what precise sense the excitations are fracton-like and in what sense they are not genuine fractons.
2- The mapped model corresponds to a transverse-field Ising chain with regularly alternating couplings. There exists relevant earlier literature on such models as for instance O. Derzhko et al., Physical Review B 66, 144401 (2002). The authors should cite and briefly discuss this and related works on the bond alternating transverse Ising chains to better place their results in the context of existing literature.
3- The observed quantum phase transition is found to belong to 2D Ising universality class, while the studied diamond spin chain is one-dimensional and maps to a transverse-field Ising chain. A short clarification that this refers to the 2D classical Ising universality class via the quantum-to-classical mapping would avoid confusion.
4- To bring a deeper insight, it would be valuable to add to the manuscript typical field dependencies of the magnetization and triplet correlation function as conjugated quantities to the field term and three-spin coupling, which would provide a further confirmation of the nature of quantum phase transition.
5- The immobile excitations are defined in terms of symmetry sectors. It would strengthen the physical interpretation if the authors briefly discuss, which local operator(s) create such an excitation and how such excitations could be detected experimentally.
6- There seem to be a few typos in equations as for instance missing brackets after summation symbols in Eqs. (8)-(9), (21) at the effective field terms, in the second and third rows in Eq. (13), and in Eq. (26).

Recommendation

Publish (easily meets expectations and criteria for this Journal; among top 50%)

---

## Editorial Decision

in_refereeing